# MVP predicts the pathogenicity of missense variants by deep learning

Hongjian Qi[1,2,9], Haicang Zhang[1,9], Yige Zhao [1,9], Chen Chen[1,3,9], John J. Long [2], Wendy K. Chung[4], Yongtao Guan[5,8] & Yufeng Shen [1,6,7✉]

Accurate pathogenicity prediction of missense variants is critically important in genetic studies and clinical diagnosis. Previously published prediction methods have facilitated the interpretation of missense variants but have limited performance. Here, we describe MVP (Missense Variant Pathogenicity prediction), a new prediction method that uses deep residual network to leverage large training data sets and many correlated predictors. We train the model separately in genes that are intolerant of loss of function variants and the ones that are tolerant in order to take account of potentially different genetic effect size and mode of action. We compile cancer mutation hotspots and de novo variants from developmental disorders for benchmarking. Overall, MVP achieves better performance in prioritizing pathogenic missense variants than previous methods, especially in genes tolerant of loss of function variants. Finally, using MVP, we estimate that de novo coding variants contribute to 7.8% of isolated congenital heart disease, nearly doubling previous estimates.

[1] Department of Systems Biology, Columbia University, New York, NY, USA. [2] Department of Applied Mathematics and Applied Physics, Columbia University, New York, NY, USA. [3] Department of Biological Sciences, Columbia University, New York, NY, USA. [4] Departments of Pediatrics and Medicine, Columbia University, New York, NY, USA. [5] Department of Pediatrics, Baylor College of Medicine, Houston, TX, USA. [6] Department of Biomedical Informatics, Columbia University, New York, NY, USA. [7] JP Sulzberger Columbia Genome Center, Columbia University, New York, NY, USA. [8] Present address: Department of Biostatistics and Bioinformatics, Duke University, Durham, NC, USA. [9] These authors contributed equally: Hongjian Qi, Haicang Zhang, Yige Zhao, Chen Chen. ✉email: ys2411@cumc.columbia.edu

Missense variants are the most common type of coding genetic variants and are a major class of genetic risk across a broad range of common and rare diseases. Previous studies have estimated that there is a substantial contribution from de novo missense mutations to structural birth defects[1–3] and neurodevelopmental disorders[4–6]. However, only a small fraction of missense de novo mutations are pathogenic[4] that will cause disease. As a result, the statistical power of detecting individual risk genes based on missense variants or mutations is limited[7]. In clinical genetic testing, many of missense variants in well-established risk genes are classified as variants of uncertain significance, unless they are highly recurrent in patients. Previously published in silico prediction methods have facilitated the interpretation of missense variants, such as CADD[8], VEST3[9], MetaSVM[10], M-CAP[11], REVEL[12], PrimateAI[13], and UNEE-CON[14]. However, based on recent de novo mutation data, they all have limited performance with low positive predictive value (Supplementary Data 1), especially in non-constrained genes (defined as ExAC[15] pLI < 0.5).

In this work, we hypothesize that missense variant pathogenicity prediction can be improved in a few dimensions. First, conventional machine learning approaches have limited capacity to leverage large amounts of training data compared to recently developed deep learning methods[16]. Second, databases of pathogenic variants curated from the literature are known to have a substantial frequency of false positives[17], which are likely caused by common issues across databases and therefore introduce inflation of benchmark performance. Developing new benchmark data and methods can help to assess and improve real performance. Finally, previous methods do not consider gene dosage sensitivity[15,18], which can modulate the pathogenicity of deleterious missense variants. Generally, missense variants can be classified into three categories based on their impact on protein function ("mode of action"): (1) gain of function (also called hypermorph or neomorph); (2) dominant negative (also called antimorph); (3) hypomorph (partial or complete loss of function, also called amorph)[19,20]. A hypomorphic variant with heterozygous genotype can only be pathogenic in dosage sensitive genes[6], whereas the pathogenicity of a gain of function or dominant negative variant with heterozygous genotype is not limited by gene dosage sensitivity. With recently published metrics of mutation intolerance, which is strongly associated with dosage sensitivity[15], it is now feasible to consider gene dosage sensitivity in predicting pathogenicity. Based on these ideas, we developed a new method, MVP, to improve missense variant pathogenicity prediction. We evaluate MVP on the data of cancer mutation hotspots and de novo variants from developmental disorders. Overall, MVP achieves better performance in prioritizing pathogenic missense variants than previous methods, especially in genes tolerant of loss of function variants.

## Results

**Model structure and predictors**. MVP uses many correlated predictors, broadly grouped into two categories (Supplementary Data 2): (a) "raw" features computed at different scales, per base pair (e.g., amino acid constraint score and conservation), per local context (e.g., protein structure and modification), as well as per gene (e.g., gene mutation intolerance, sub-genic regional depletion of missense variants[21]); (b) deleteriousness scores from selected previous methods. To account for the impact of gene dosage sensitivity on pathogenicity of heterozygous missense variants, we trained our models in constrained genes (defined as ExAC[15] probability of being loss-of-function (LoF) intolerant (pLI) ≥ 0.5) and non-constrained genes (ExAC pLI < 0.5) separately. We included 38 features for the constrained gene model, and 21 features for the non-constrained gene model. We removed most of published prediction methods features due to limited prediction accuracy (Supplementary Data 1, 2). We also excluded most of conservation features in the non-constrained gene model, as pathogenic variants in non-constrained genes are less conserved (Supplementary Fig. 1).

MVP uses a deep residual neural network (ResNet)[22] model. ResNet was designed for computer vision[22] and was successfully applied in structural biology[23] and genomics[24]. The convolutional layers in ResNet are capable of extracting hierarchical features or nonlinear spatial local patterns from images or sequence data. To take advantage of this, we ordered the predictors based on their correlation, as highly correlated predictors are clustered together (Supplementary Fig. 2). There are two layers of residual blocks, consisting of convolutional filters and activation layers, and two fully connected layers with sigmoid output (Supplementary Fig. 3). For each missense variant, we defined MVP score by the rank percentile of the ResNet's raw sigmoid output relative to all 76 million possible missense variants.

In this work, we focus on the rare variants with large effect. We used a minor allele frequency (MAF) threshold of $10^{-4}$ (based on gnomAD[25]) to filter variants in both training and testing data sets.

**Model training**. We obtained large curated data sets of pathogenic variants from HGMD[26] and UniProt[10,27] as positives and random rare missense variants from population data as negatives for training (Supplementary Data 3). The training process takes around 10 min on 1.6 GHz GPU with 2560 cores and 8 GB memory. Using 6-fold cross-validation on the training set (Supplementary Fig. 4), MVP achieved mean area under the curve (AUC) of 0.99 in constrained genes and 0.97 in non-constrained genes.

**Performance evaluation using curated mutation data sets**. To evaluate predictive performance of the MVP and compare it with other methods, we obtained an independent curated testing data set from VariBench[10,28] (Supplementary Fig. 5). MVP outperformed all other methods with an AUC of 0.96 and 0.92 in constrained and non-constrained genes, respectively. A few recently published methods (REVEL, M-CAP, VEST3, and MetaSVM) were among the second-best predictors and achieved AUC around 0.9.

Similar to HGMD and Uniprot data used in training, VariBench data are curated from literature. False positives caused by similar factors by this approach across training and VariBench data sets could inflate the performance in testing. To address this issue, we compiled cancer somatic mutation data for further evaluation, including missense mutations located in inferred hotspots based on statistical evidence from a recent study[29] as positives, and randomly selected variants from DiscovEHR[30] database as negatives. In this dat aset, the performance of all methods decreased, but MVP still achieved the best performance with AUC of 0.91 (maximum $p = 3.2e-8$) and 0.85 (maximum $p = 6.6e-4$) in constrained and non-constrained genes, respectively (Fig. 1). We observed that methods using HGMD or UniProt in training generally have greater performance drop than others (Supplementary Data 1, Supplementary Fig. 6, Supplementary Note 1).

To investigate the contribution of features to MVP predictions, we performed cross-one-group-out experiments and used the differences in AUC as an estimation of feature contribution (Fig. 2). We found that in constrained genes, conservation scores and published deleteriousness predictors have relatively large contribution, whereas in non-constrained genes, protein structure

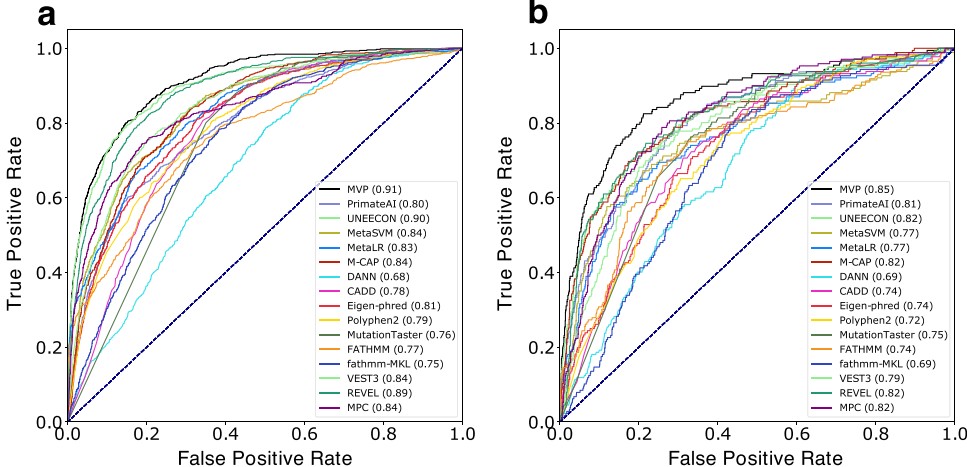

**Fig. 1 Comparison of receiver operating characteristic (ROC) curves of existing prediction scores and MVP scores using cancer somatic mutation hotspot data. a** Constrained genes (ExAC pLI ≥ 0.5): evaluation of 698 cancer mutations located in hotspots from 150 genes, and 6989 randomly selected mutations from DiscovEHR database excluding mutations used in training. **b** Non-constrained genes (ExAC pLI < 0.5): evaluation of 177 cancer mutations located in hotspots from 55 genes and 1782 randomly selected mutations from DiscovEHR database excluding mutations used in training. The performance of each method is evaluated by the ROC curve and area under the curve (AUC) indicated in parenthesis. Higher AUC score indicates better performance.

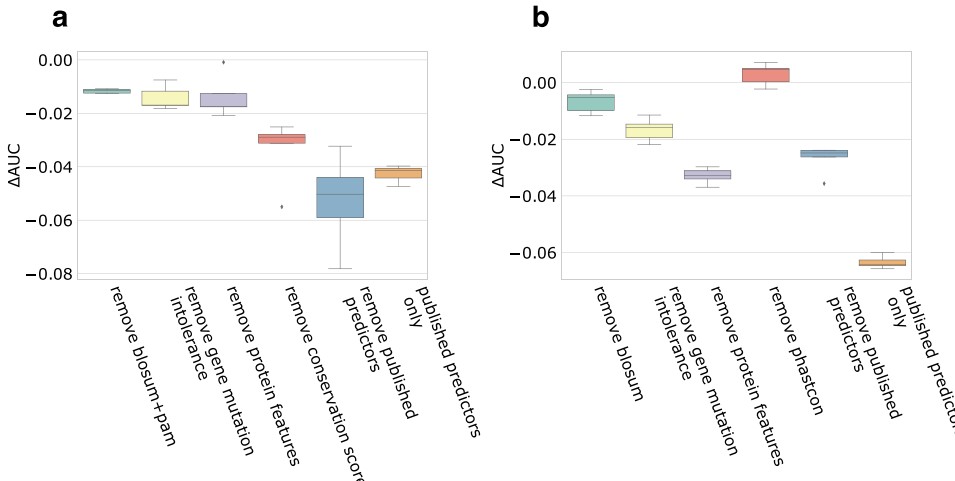

**Fig. 2 Measuring the contribution of features to MVP prediction performance in cancer mutation hotspots data.** Performance contribution is measured by AUC reduction (ΔAUC) from excluding a group of features. Since features within a group is often highly correlated, we did measure the contribution of an entire group instead of individual features in the group. **a** Constrained genes (**b**) Non-constrained genes. We subsampled negatives from DiscovEHR database and calculated ΔAUC 15 times and estimated the error bar. The box bounds the interquartile range (IQR) divided by the median, and Tukey-style whiskers extend to a maximum of 1.5× IQR beyond the box.

and modification features and published predictors are the most important.

**Evaluating pathogenic de novo missense variants in CHD and ASD.** To test the utility in real genetic studies, we obtained germline de novo missense variants (DNMs) from 2645 cases in a congenital heart disease (CHD) study[2] (Supplementary Data 4), 3953 cases in autism spectrum disorder (ASD) studies[2,4,5] (Supplementary Data 5), and DNMs from 1911 controls (unaffected siblings) in Simons Simplex Collection[2,4,5] (Supplementary Data 6). Since genes with cancer mutation hotspots are relatively well studied in both constrained and non-constrained gene sets, assessment using de novo mutations can provide additional insight with less bias (Supplementary Data 7). Because the true pathogenicity of most of the de novo mutations is unknown, we cannot directly evaluate the performance of prediction methods. To address this issue, we first compared the distribution of predicted scores of DNMs in cases with the ones in controls (Fig. 3).

Using Mann–Whitney U test, MVP-predicted scores of variants in cases and controls are significantly different ($p = 1e−5$ and $2.7e−4$ for CHD vs. controls and ASD vs. controls, respectively), and the difference is greater than predictions from other methods.

We then calculated the enrichment rate of predicted pathogenic DNMs by a method with a certain threshold in the cases compared to the controls, and then estimated precision and the number of true risk variants (Methods), which is a proxy of recall since the total number of true positives in all cases is a (unknown) constant independent of methods. We compared the performance of MVP to other methods by estimated precision and recall-proxy (Fig. 4). Based on the optimal thresholds of MVP in cancer hotspot ROC curves, we used a score of 0.7 in constrained genes and 0.75 in non-constrained genes to define pathogenic DNMs (Supplementary Fig. 7). In constrained genes, we observed an enrichment of 2.2 in CHD and an enrichment of 1.9 in ASD (Supplementary Data 8, 9), achieving estimated precision of 0.55 and 0.47 (Fig. 4a and 4d), respectively. This indicates that about

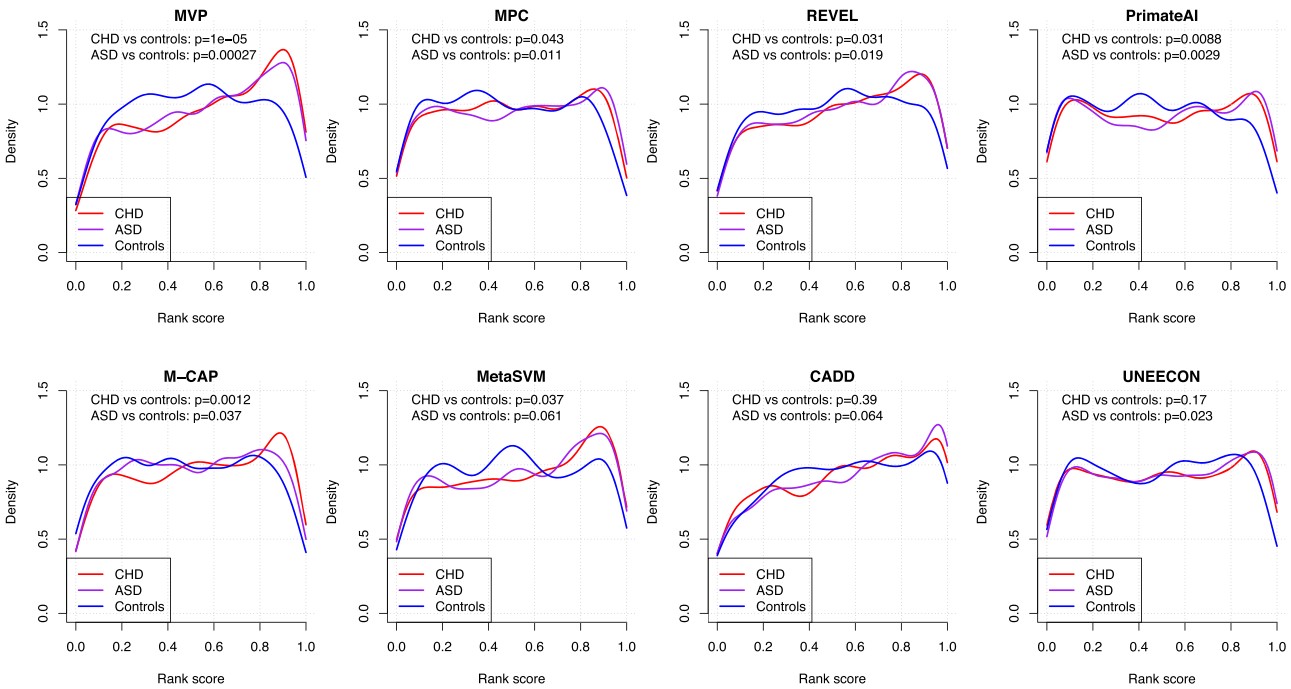

**Fig. 3 Distribution of predicted scores of de novo missense variants by MVP and other methods.** For each method, we normalized all predictions by rank percentile, and used two-sided Mann–Whitney U test to assess the statistical significance of the difference between cases and controls. CHD: congenital heart disease; ASD: autism spectrum disorder; controls: unaffected siblings from the ASD study. Number of de novo missense variants compared: CHD: 1486; ASD: 2050; controls: 838.

50% of the MVP-predicted pathogenic DNMs contribute to the diseases. In non-constrained genes, we observed an enrichment of 1.5 in CHD and 1.4 in ASD (Supplementary Data 8, 9), respectively, and 0.33 and 0.29 in estimated precision (Fig. 4b and 4e). In all genes combined, MVP achieved an estimated precision of 40% for CHD and 34% for ASD (Fig. 4c and 4f). The next best methods reached 26% (M-CAP[11]) and 21% (Prima-teAI[13]) given the same recall-proxy for CHD and ASD, respectively (Supplementary Data 8, 9). Furthermore, the estimated precision of MVP with DNMs at optimal threshold is much closer to the expected precision based on ROC of cancer hotspots data than the value from VariBench data (Supplementary Fig. 8 and Supplementary Note 1), supporting that there is less performance inflation in testing using cancer data.

We used ResNet in MVP, which is based on Convolutional Neural Network while more efficient in training. In order to assess its importance, we trained a Random Forest model, which is the core of methods such as REVEL[12], with the same training sets and features. We also trained a Fully-connected Neural Network as the baseline performance of neural network methods. ResNet slightly outperformed Random Forest and has the same performance with Fully-connected Neural Network with cancer somatic mutation data (Supplementary Fig. 9), and substantially outperformed both with de novo mutations in CHD and autism (Supplementary Figs. 10, 11).

Previous studies have estimated that deleterious de novo coding mutations, including loss of function variants and damaging missense variants, have a small contribution to isolated CHD[2]. Here, we used MVP to revisit this question. With the definition of damaging DNMs in Jin et al. 2017[2] (based on MetaSVM[10]), the estimated contribution of deleterious de novo coding mutations to isolated CHD is about 4.0%. With MVP score of 0.75, the estimation is 7.8% (95% CI = [5.9%, 9.6%]), nearly doubling the previous estimate (Supplementary Data 10, 11). We performed pathway enrichment analysis of genes with MVP-predicted pathogenic de novo missense mutations in CHD

cases using Enrichr[31]. In isolated cases, the genes with such variants are significantly enriched (FDR < 0.01) for cardiac conduction (FDR = 0.006, odds ratio = 8.7) and muscle contraction (FDR = 0.006, odds ratio = 6.8). In syndromic CHD cases who have additional congenital anomalies or neurodevelopmental disorders, the genes with such variants are significantly enriched in Notch, Robo, or MAPK signaling pathways (Supplementary Fig. 12, Supplementary Data 12) that have been implicated with other developmental disorders.

## Discussion

We developed a new method, MVP, to predict pathogenicity of missense variants. MVP is based on residual neural networks, a supervised deep learning approach, and was trained using a large number of curated pathogenic variants from clinical databases, separately in constrained genes and non-constrained genes. Using cancer mutation hotspots and de novo mutations from CHD and ASD studies, we showed that MVP achieved overall better performance than published methods, especially in non-constrained genes.

Two factors may contribute to the improved prediction performance. First, deep neural network models have a larger learning capacity to leverage large training data set than conventional machine learning methods that were used in previous publications. The residual blocks in ResNet largely alleviate the problem of vanishing gradient when increasing the number of layers, which further enables efficient training on deeper networks[22]. Using the same features in training data set, we showed that ResNet achieved better performance than two conventional machine learning methods, Random Forest and Fully-connected Neural Network. Second, we trained the model separately in constrained and non-constrained genes. This is motivated by the idea that the mode of action of pathogenic variants can be different in constrained and non-constrained genes. Constrained genes are likely to be dosage sensitive, therefore, deleterious

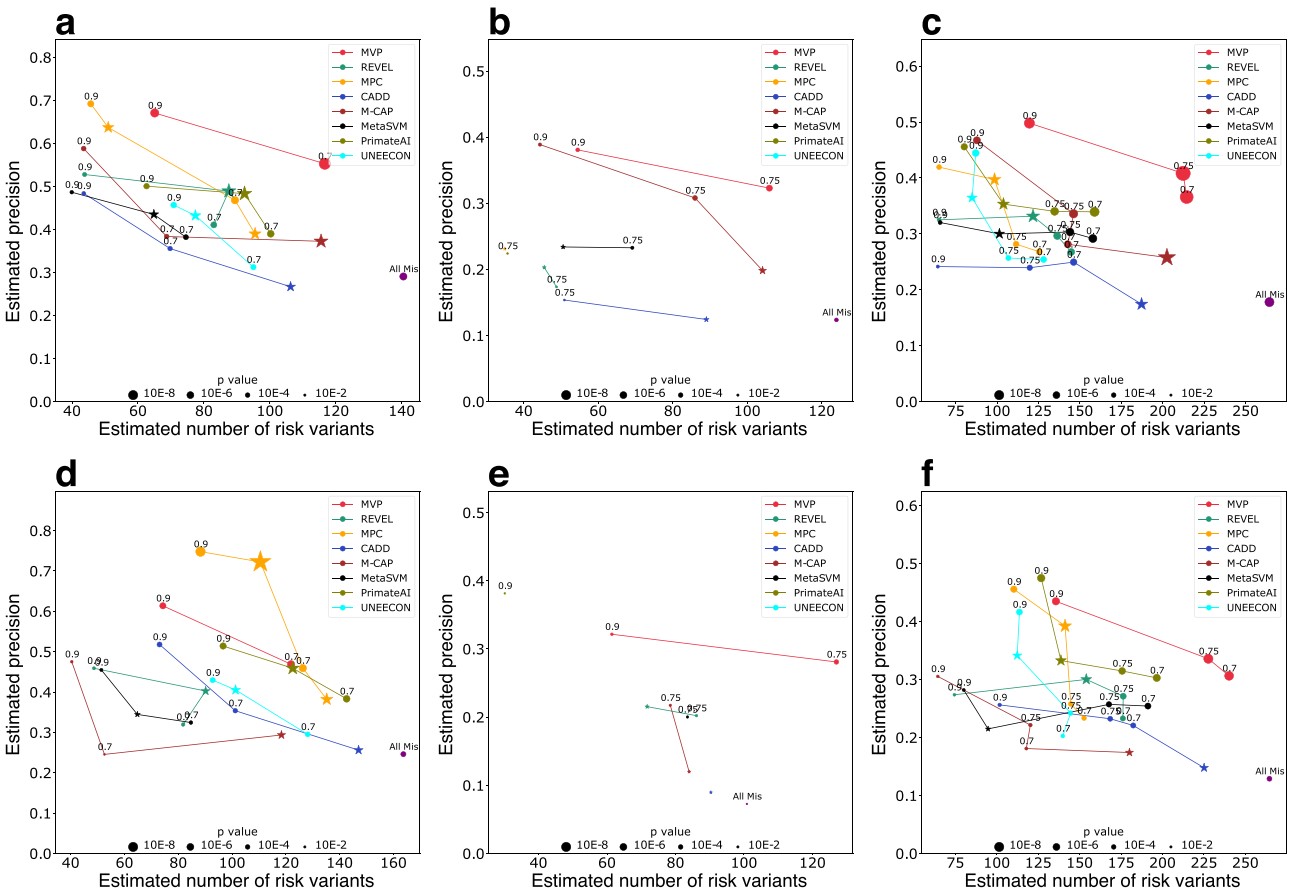

**Fig. 4 Comparison of MVP and previously published methods using de novo missense mutations from CHD and ASD studies by precision-recall-proxy curves.** Numbers on each point indicate rank percentile thresholds, star points indicate thresholds recommended by publications. The positions of "All Mis" points are estimated from all missense variants in the gene set without using any pathogenicity prediction method. The point size is proportional to −log (p-value). P-value is calculated by two-sided binomial test, only points with p value less than 0.05 are shown. **a–c** Performance in CHD DNMs in constrained genes, non-constrained genes, and all genes, respectively. **d–f** Performance in ASD DNMs in constrained genes, non-constrained genes, and all genes, respectively.

missense variants of any mode of action with heterozygous genotypes can be pathogenic. In contrast, non-constrained genes are unlikely to be dosage sensitive, as a result, hypermorphic variants with heterozygous genotype are unlikely to be pathogenic. Instead, gain of function or dominant negative variants are the main type of pathogenic variants in these genes. By training the model in these two groups of genes separately, we enhance the ability of the model to use features that are informative with gain of function or dominant negative in non-constrained genes. In addition, we selected a smaller number of features in the model for non-constrained genes. We excluded most of the conservation scores and published predictors, based on the observation that known pathogenic variants in non-constrained genes are less conserved across species (Supplementary Fig. 1) and that published methods have poor performance in prioritization of de novo variants in these genes (Supplementary Data 1). We kept all the features related to protein structure and post-translational modification. The change of the model for non-constrained genes is supported by sensitivity analysis using cancer mutation hotspot data (Fig. 2). When protein structure related features are further excluded, the model has the biggest drop in AUC in non-constrained genes. In contrast, excluding conservation scores does not lead to reduced AUC.

In a recent genetic study of CHD[1,2], de novo mutations were estimated to contribute to 10–30% of syndromic cases who had additional congenital anomalies or neurodevelopmental disorders

(NDD), but only about 4% of isolated cases who did not have additional anomalies or NDD. Isolated CHD are the most common form of the disease[32]. We reanalyzed the de novo missense variants using MVP. We estimated that predicted-pathogenic de novo mutations actually contribute to about 7.8% of isolated cases, doubling previous estimate. The revised estimate suggests a greater utility of discovery or prioritization of new candidate risk genes by de novo mutations in isolated CHD. Pathway analysis shows predicted pathogenic missense variants in isolated CHD cases are enriched in cardiac conduction and muscle contraction, whereas the ones in syndromic CHD cases are enriched in Notch, Robo, or MAPK signaling. The different organ-specificity of these pathways is consistent with phenotypes. Nevertheless, the genetic cause of CHD is very heterogeneous[1,2], similar to autism[4,5,33–35]. With a few thousand cases, very few true risk genes have multiple observed deleterious de novo mutations in the data. As a result, there is a substantial uncertainty of the estimated contribution of de novo mutations. Future studies with larger sample size will enable more precise estimation of the contribution of de novo mutations and identification of new candidate risk genes.

We note that MPC has better precision than MVP with constrained genes with ASD data (Fig. 4d), while MVP has much better performance in constrained genes with CHD data (Fig. 4a), and much better performance in non-constrained genes with both ASD data (Fig. 4e) and CHD data (Fig. 4b). Based on known risk genes for ASD[36] and CHD[2], autism risk genes are under

stronger negative selection of loss of function variants ("s_het"[37]) than CHD risk genes. High score of sub-genic regional constraint, a key metric used in MPC, is predominantly located in genes under very strong negative selection. This could explain the difference of autism vs. CHD mutations among constrained genes. Supplementary Fig. 13 shows that variants in constrained genes significantly have higher MPC scores than those in non-constrained genes, either for pathogenic variants or benign variants in the training set. MPC has limited performance among non-constrained genes, the results of all threshold in MPC in non-constrained genes is not statistically significant in ASD cases, and MPC was not shown in the panel on non-constrained genes (Fig. 4d and Fig. 4e). We also noted that MPC achieves better sensitivity at the published recommended threshold compared to MVP with all genes with CHD data (Fig. 4c). There is a trade-off between sensitivity and specificity. The recommended threshold of MPC achieves slightly better sensitivity compared to recommended threshold of MVP, however, the specificity of MVP is almost doubling MPC. With the same sensitivity, MVP generally outperformed other methods.

Pathogenic variants may have a range of effect size and penetrance. In this work, we focused on rare variants with large effect size. Our model is consistent with a common definition of pathogenicity[38], that is, pathogenic variants mechanistically contribute to disease, but not necessarily with full penetrance. Specifically, we included prediction features based on protein structure and protein modification, which provide biophysical and biochemical basis for mechanistic contribution to the disease, and evolution conservation, which is usually a consequence of direct contribution to diseases. In addition, the positives in model training were from HGMD and UniProt, expert-curated databases with large-number of likely pathogenic and rare variants reported in previous publications. We do not include disease-associated common variants from genome-wide association studies (GWAS) in training.

A limitation of MVP is the unknown but potentially high false positive rate of curated pathogenetic variants used in training. The largest databases, such as HGMD and Varibench, were curated from a broad range of publications with uneven quality. As a result, independently curated databases suffer from similar types of errors. This issue is highlighted by the performance drop in testing on cancer somatic mutation hotspots data comparing to VariBench. Systematic efforts such as ClinVar[39] will eventually produce better and larger training data to improve prediction performance.

Finally, we note that a single pathogenic score cannot capture the complexity of the mechanisms of pathogenicity. Many genes may have different modes of action in different diseases. One example is CTNNB1 (beta catenin). It is an oncogene in cancer through gain-of-function mutations[40], and a risk gene in structural birth defects and neurodevelopmental disorders through loss of function germline mutations[41]. Most prediction tools, including MVP, do not distinguish pathogenic missense variants with gain of function from the ones with loss of function. Explicitly predicting gain or loss of function, as what a recent study focused on channels did[42], would improve the utility of prediction methods.

## Methods
**Training data sets**. We compiled 22,390 missense mutations from Human Gene Mutation Database Pro version 2013 (HGMD)[26] database under the disease mutation (DM) category, 12,875 deleterious variants from UniProt[10,27], and 4424 pathogenic variants from ClinVar database[39] as true positive (TP). In total, there are 32,074 unique positive training variants. The negative training sets include 5190 neutral variants from Uniprot[10,27], randomly selected 42,415 rare variants from DiscovEHR database[30], and 39,593 observed human-derived variants[8]. In total, there are 86,620 unique negative training variants (Supplementary Data 3).

**Testing data sets**. We have three categories of testing data sets (Supplementary Data 3). The three categories are: (a) Benchmark data sets from VariBench[10,28] as positives and randomly selected rare variants from DiscovEHR database[30] as negatives; (b) cancer somatic missense mutations located in hotspots from recent study[29] as positives and randomly selected rare variants from DiscovEHR database[30] as negatives; (c) de novo missense mutation data sets from recent published exome-sequencing studies[2,4,5]. All variants in (a) and (b) that overlap with training data sets were excluded from testing.

We tested the performance in constrained genes (ExAC pLI ≥ 0.5) and non-constrained gene (ExAC pLI < 0.5)[15] separately.

To focus on rare variants with large effect, we selected ultra-rare variants with $MAF < 10^{-4}$ based on gnomAD database to filter variants in both training and testing data sets. We applied additional filter of $MAF < 10^{-6}$ for variants in constrained genes in both cases and controls for comparison based on a recent study[21,43].

**Features used in the MVP model**. MVP uses many correlated features as predictors (Supplementary Data 2). There are six categories: (1) local context: GC content within 10 flanking bases on the reference genome; (2) amino acid constraint, including blosum62[44] and pam250;[45] (3) conservation scores, including phyloP 20way mammalian and 100way vertebrate[46], GERP++[47], SiPhy 29way[48], and phastCons 20way mammalian and 100way vertebrate;[49] (4) Protein structure, interaction, and modifications, including predicted secondary structures[50], number of protein interactions from the BioPlex 2.0 Network[51], whether the protein is involved in complexes formation from CORUM database[52], number of high-confidence interacting proteins by PrePPI[53], probability of a residue being located the interaction interface by PrePPI (based on PPISP, PINUP, PredU), predicted accessible surface areas were obtained from dbPTM[54], SUMO scores in 7-amino acids neighborhood by GPS-SUMO[55], phosphorylation sites predictions within 7 amino acids neighborhood by GPS3.0[56], and ubiquitination scores within 14-amino acids neighborhood by UbiProber;[57] (5) Gene mutation intolerance, including ExAC metrics[15] (pLI, pRec, lof_z) designed to measure gene dosage sensitivity or haploinsufficiency, RVIS[58], probability of causing diseases under a dominant model "domino"[59], average selection coefficient of loss of function variants in a gene "s_het"[37], and sub-genic regional depletion of missense variants;[21] (6) Selected deleterious or pathogenicity scores by previous published methods obtained through dbNSFPv3.3a[60], including Eigen[61], VEST3[9], MutationTaster[62], PolyPhen2[63], SIFT[64], PROVEAN[65], fathmm-MKL[66], FATHMM[66], MutationAssessor[67], and LRT[68].

For consistency, we used canonical transcripts to define all possible missense variants[21]. Missing values of protein complex scores are filled with 0 and other features are filled with −1.

Since pathogenic variants in constrained genes and non-constrained genes may have different mode of action, we trained our models on constrained and non-constrained variants separately with different sets of features (38 features used in constrained model, 21 features used in non-constrained model, Supplementary Data 2).

**Deep learning model**. MVP is based on a deep residual neural network model (ResNet)[22] for predicting pathogenicity using the predictors described above. To preserve the structured features in training data, we ordered the features according to their correlations (Supplementary Fig. 3). The model (Supplementary Fig. 2) takes a vector of the ordered features as input, followed by a convolutional layer of 32 kernels with size 3 × 1 and stride of 1, then followed by 2 computational residual units, each consisting of 2 convolutional layers of 32 kernels with size 3 × 1 and stride of 1 and a ReLU[69] activation layer in between. The output layer and input layer of the residual unit is summed and passed on to a ReLU activation layer. After the two convolutional layers with residual connections, 2 fully connected layers of 320 × 512 and 512 × 1 are used followed by a sigmoid function to generate the final output[70].

$$\text{Sigmoid}(x) = \frac{1}{1 + e^{-x}}$$

(Supplementary Fig. 2). In training, we randomly partitioned the synthetic training data sets into two parts, 80% of the total training sets for training and 20% for validation. We trained the model with batch size of 64, used adam[71] optimizer to perform stochastic gradient descent[72] with cross-entropy loss between the predicted value and true value. After one full training cycle on the training set, we applied the latest model weights on validation data to compute validation loss.

To avoid over fitting, we used early stopping regularization during training. We computed the loss in training data and validation data after each training cycle and stopped the process when validation loss is comparable to training loss and do not decrease after 5 more training cycle, and then we set the model weights using the last set with the lowest validation loss. We applied the same model weights on testing data to obtain MVP scores for further analysis.

**Hyperparameters in MVP**. In the MVP neural network, we tested different number of residual blocks for the model structure. With all other parameters fixed, the model with two residual blocks contain 12,544 parameters before fully connected layers, and it saturates at around 20 iterations. Deeper models lead to fast overfitting and unstable performance in testing data sets (Supplementary Fig. 19).

We used AUROC on cancer hotspot data to investigate the effects of the number of residual blocks on the testing performance. When 8 and 16 residual blocks are used, the AUROC for constrained genes are 0.885 and 0.872, respectively, and the AUROC for non-constrained genes are 0.822 and 0.814, respectively. Other hyperparameters we chose are commonly used in deep learning models, including kernel size of 3, pooling size of 2, filter size of 32 and ReLU as activation functions.

**Previously published methods for comparison**. We compared MVP score to 13 previously published prediction scores, namely, M-CAP[11], DANN[73], Eigen[61], Polyphen2[63], SIFT[64], MutationTaster[62], FATHMM[66], REVEL[12], CADD[8], MetaSVM[10], MetaLR[10], VEST3[9], and MPC[21].

**Normalization of scores using rank percentile**. For each method, we first obtained predicted scores of all possible rare missense variants in canonical transcripts, and then sort the scores and converted the scores into rank percentile. Higher rank percentile indicates more damaging, e.g., a rank score of 0.75 indicates the missense variant is more likely to be pathogenic than 75% of all possible missense variants.

**ROC curves**. We plotted Receiver operating characteristic (ROC) curves and calculated Area Under the Curve (AUC) values in training data with 6-fold cross validation (Supplementary Fig. 4), and compared MVP performance with other prediction scores in curated benchmark testing data sets (Supplementary Fig. 5) and cancer hotspot mutation data set (Fig. 1). For each prediction method, we varied the threshold for calling pathogenic mutations in a certain range and computed the corresponding sensitivity and specificity based on true positive, false positive, false negative and true negative predictions. ROC curve was then generated by plotting sensitivity against 1—specificity at each threshold.

**Optimal points based on ROC curves**. We define the optimal threshold for MVP score as the threshold where the corresponding point in ROC curve has the largest distance to the diagonal line (Supplementary Fig. 7). Based on the true positive rate and false positive rate at the optimal points in ROC curves, we can estimate the precision and recall in de novo precision-recall-proxy curves (Supplementary Fig. 8 and Supplementary Note 1).

**Precision-recall-proxy curves**. Since de novo mutation data do not have ground truth, we used the excess of predicted pathogenic missense de novo variants in cases compared to controls to estimate precision and proxy of recall. For various thresholds of different scores, we can calculate the estimated number of risk variants and estimated precision based on enrichment of predicted damaging variants in cases compared to controls. We adjusted the number of missense de novo mutation in controls by the synonymous rate ratio in cases verses controls, assuming the average number of synonymous as the data sets were sequenced and processed separately) (Supplementary Table 2), which partly reduced the signal but ensures that our results were not inflated by the technical difference in data processing.

Denote the number of cases and controls as $N_1$ and $N_0$, respectively; the number of predicted pathogenic de novo missense variants as $M_1$ and $M_0$, in cases and controls, respectively; the rate of synonymous de novo variants as $S_1$ and $S_0$, in cases and controls, respectively; technical adjustment rate as $\alpha$; and the enrichment rate of variants in cases compared to controls as $R$.

We first estimate $\alpha$ by:

$$\alpha = \frac{S_1}{S_0} \tag{1}$$

Then assuming the rate of synonymous de novo variants in cases and controls should be identical if there is no technical batch effect, we use $\alpha$ to adjust estimated enrichment of pathogenic de novo variants in cases compared to the controls by:

$$R = \frac{\frac{M_1}{N_1}}{\frac{M_0}{N_0} \times \alpha} \tag{2}$$

Then we can estimate number of true pathogenic variants ($M_1'$) by:

$$M_1' = \frac{M_1(R-1)}{R} \tag{3}$$

And then precision by:

$$\widehat{\text{Precision}} = \frac{M_1'}{M_1} \tag{4}$$

**Random forest model and fully-connected neural network model**. To analyze the relative contribution of the features and the deep neural network to the improvement over exiting methods, we trained a Random Forest (RF) model and Fully-connected Neural Network (FCNN) model with the same features as our base line. RF and FCNN were implemented using Python package scikit-learn and Keras separately. We tried several hyperparameter settings of RF and FCNN, and the best settings were determined by 10-fold cross-validation. In our RF model, we used 256 trees with depth of 12. The minimum number of examples per node that were allowed to be split was set as 10 to alleviate overfitting. For FCNN, we used 2 hidden layers with 64 neurons and 32 neurons separately and ReLU as the activation function. We used Adam algorithm to minimize the objective function with hyperparameters lr = 1e−4, β = 0.99, and ε = 1e−8. Early-stopping was applied with validation error as the metric. The 10 trained models were all kept, and for testing, we average the outputs of the 10 models as the final predicted scores.

**Reporting summary**. Further information on research design is available in the Nature Research Reporting Summary linked to this article.

## Data availability

Precomputed MVP pathogenicity score for all possible missense variants in canonical transcripts on human hg19 can be downloaded from: https://figshare.com/articles/dataset/Predicting_pathogenicity_of_missense_variants_by_deep_learning/13204118. All other data supporting the findings of this study are available within the paper and its Supplementary information files. The training and testing data can be accessed through http://www.discovehrshare.com/downloads (DiscovEHR), http://structure.bmc.lu.se/VariBench/substitutionsall.php (VariBench), http://www.hgmd.cf.ac.uk/ac/index.php (HGMD), https://www.uniprot.org/docs/humpvar (UniProt), and https://ftp.ncbi.nlm.nih.gov/pub/clinvar/vcf_GRCh37/ (ClinVar).

## Code availability

Python scripts used for model training and testing are available on GitHub: https://github.com/ShenLab/missense.

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

## Acknowledgements

We thank Na Zhu, Ben Lai, Itsik Pe'er, Emily Gao, and Jiayao Wang for helpful discussions. We thank Pediatric Cardiac Genomics Consortium (PCGC) investigators for CHD data access, and the patients and their families for their generous contribution to the PCGC study and to the Simons Simplex Collection study families and principal investigators (A.L. Beaudet, R. Bernier, J. Constantino, E.H. Cook, Jr, E. Fombonne, D. Geschwind, D.E. Grice, A. Klin, D.H. Led-better, C. Lord, C.L. Martin, D.M. Martin, R. Maxim, J. Miles, O. Ousley, B. Peterson, J. Piggot, C. Saulnier, M.W. State, W. Stone, J.S. Sutcliffe, C.A. Walsh, and E. Wijsman) and the coordinators and staff at the SSC clinical sites. This work was supported by NIH grants R01GM120609 (H.Q., H.Z, W.K.C., and Y.S.), U01HG008680 (Y.S.), U01HL098163 (W.K.C. and Y.S.), P30DK026687 (W.K.C.), and Simons Foundation (W.K.C.).

## Author contributions

All authors contributed to data analysis, interpretation, and manuscript writing. Y.S. conceived and designed the study.

## Competing interests

The authors declare no competing interests.
