## [Peer Review File · Nature Communications]

Reviewers' comments:

Reviewer #1 (Remarks to the Author):

In this manuscript, the authors describe a new deep learning method called MVP for predicting pathogenicity of missense variants. Based on several different data sets, the proposed method was shown to perform clearly better than the other methods considered in the comparisons. Whether the better performance can be attributed to the new method requires additional supporting data, as detailed below.

Major comments:

1. The authors have focused on missense variants only. For many complex diseases, the statistically associated coding variants have been well-cataloged and/or they can only explain a certain (sometimes fairly small) fraction of disease susceptibility/heritability. The authors have chosen not to focus on the much larger set of non-coding variants and the more difficult problem of predicting effects of non-coding variants. Even for genic variants, the authors have chosen not to consider nonsense, frameshift and splicing variants. Considering that in disease genomic studies the significance of a coding variant is indicated by its relative frequencies in cases and controls, the primary use of the proposed method is in the study of rare variants. Given all these restrictions, the authors need to explain more clearly the value of their method in terms of the magnitude of increased accuracy as compared to other previous methods.

2. The manuscript does not include a clear definition of pathogenicity, but implicitly implies its meaning based on the data sets used (by adopting the definitions used by these data sets) or recurrence in cases and controls. Since pathogenic phenotype could be defined in various ways covering a wide spectrum of severity, it is important to explain which definition(s) is(are) used in this study and whether the data sets considered are consistent with this(these) definition(s).

3. The pathogenicity of a variant should depend highly on the particular disease of interest. For instance, if a variant disrupts an important function of a protein, but the protein is not expressed in the relevant cell or tissues types of a disease, it may not be pathogenic in the context of this disease. In the whole manuscript, pathogenicity is discussed in a way that does not take into account the disease context. This is exemplified by the training of a single model that was applied to several different data sets for very different diseases. Is it sufficient to construct a single general model for all diseases?

4. The authors suggest that MVP performs better than the previous methods because of the methodology, including the use of deep residual network and the choice of input features. This could indeed be the case, but it is also possible that the method performed better due to the training data. There are two related issues here:

A. Is it possible to train some of the previous methods using exactly the same training set? In particular, if the CNN method DANN is trained using this set of data, would it also perform as well as MVP? Without this controlled comparison, it is hard to evaluate the significance of the proposed deep residual network architecture.

B. The authors have carefully removed variants in the testing sets that are also present in the training set. However, it is not clear whether some features directly contained information of the testing variants. In particular, since the predictions of many methods were collected from dbNSFP and they were used as input features to MVP, it is important to ensure that the testing data were not involved in training any of these methods.

5. Related to #4a above, the authors have only cited a reference and given an abstract explanation (that deep residual network can better use big training data) for their choice of deep residual network over other machine learning methods. A more thorough discussion would be necessary to help explain the advantages of this method and guide future research on similar topics.

6. By predicting the pathogenicity of each variant independently, the authors are making an assumption that the disease phenotype does not depend on the interaction of multiple variants.

7. While the authors have considered many types of features, it seems that they have not (at least directly) use features related to protein domains, functional annotations of genes and context-specific activities such as gene expression.

8. In Supplementary Table 1, how are the score thresholds of the different methods determined? If the objective is to maximize the precision alone, usually it can be achieved by setting a higher score threshold to predict a smaller number of risk variants.

9. To what extent is it confident that the random rare missense variants from population data can be considered negative training data?

10. The negative data used in defining the testing sets need to be explained in more detail. This is important for evaluating whether there are some obvious differences between the positives and negatives.

11. In Figure S6a, it is seen that for constrained genes, removing the published predictors led to a fairly big decrease of the AUC value. What would be the AUC value with only these predictors, integrated by the proposed deep residual network or a simpler meta-learning method?

12. Most of the performance evaluations in this study are based on ranking of the predicted pathogenicity. This decision of the authors is reasonable, since the issue of finding a suitable score threshold for every method can be avoided. On the other hand, the absolute score could also carry important information. For instance, practically users of these scores may want to focus on the variants considered "severe" or "damaging" by the prediction method, rather than focusing on a fixed number of top variants. Did the authors find a particular score threshold/range of their method that seems suitable for the different data sets considered, or do they have a recommended way for tuning the threshold when analyzing an unseen data set?

Minor comment:

1. In Figure S3, how can the mean of six 0.99 to be 0.98?

Reviewer #2 (Remarks to the Author):

Qi et al. describes a new method for predicting pathogenic missense mutations called MVP. Compared to previous methods, MVP uses deep residual neural network to leverage large training data sets and to deal with correlated co-variables. Using several large sets of annotated pathogenic missense mutations, the authors demonstrate MVP outperforms several state-of-the art methods. The use of deep neural network is innovative although all statistical features used in MVP have been tested before. Given the black-box nature of deep neural network, it is hard to figure out the underlying mechanisms that lead to the performance gain. Additionally, the following major concerns need to be addressed before the manuscript becomes suitable for publication.

Major comments:

1. "We reason that the variants in constrained genes (ExAC pLI \geq 0.5) and non-constrained gene (ExAC pLI $<$ 0.5) may have different modes of action of pathogenicity"?

2. Can the authors provide more rationale and supporting evidence for this hypothesis? How would the performance results change if a difference pLI threshold is chosen?

3. A sensitivity analysis of the deep neural network model needs to be performed, i.e. how does the various parameter setting of the neural network affect its performance?

4. AUC curves are used for performance comparison. However, given the small difference in the AUC scores, are they statistically significant? For instance, in Figure 1, is the difference between MVP and REVEL statistically significant? And the difference between MVP and M-CAP and REVEL significant? Same question for the supplemental figures.

5. There are a few issues with Figure 3. Figure 3D shows MPC outperforms MVP in ASD constrained genes. Any explanation? In Figure 3E, why there is no curve for MPC? In Figure 3C for all genes, doesn't MPC achieves better sensitivity at the published recommended threshold (indicated by a star) compared to MVP (threshold = 0.7)?

6. The manuscript is written without enough details. For instance, how are previous methods run in the benchmarking analysis? Using default parameters? The Discussion section is not really a discussion. It should be expanded to discuss issues such as 1) what is the reason for the performance gain by the deep neural network approach; 2) Additional candidate pathogenic mutations are predicted for CHD. But what are they, any new biological insights?

Reviewer #3 (Remarks to the Author):

The manuscript "MVP: predicting pathogenicity of missense variants by deep learning" describes a new method to predict pathogenicity of missense variants using a deep learning approach, with convolutional neural network (CNN) called a residual network (ResNet). The study is timely and the effort of applying cutting-edge machine learning approaches in pathogenicity prediction are important. However, I have several major concerns regarding the study design and the results.

1. As a neutral/benign dataset, the authors could use a set of random variants from the human population (such as from gnomAD), since such set will contain by definition a vast proportion of variants that are not pathogenic and represent the average human exome.

2. The negative training set consists of rare variants from DiscovEHR, Uniprot negatives and Human-derived changes from the CADD database. It is important that the authors specify what is the MAF range and CADD scores of mutations employed from each of these three datasets. The concern is that the authors are comparing highly pathogenic mutations against trivially benign variants which are excluded as a first step of data filtering in any WES study. The other concern is training on CADD database mutations when CADD itself is one of the training features.

3. The authors perform thorough evaluations and show very good performance of their method when comparing it with 13 other pathogenicity prediction tools that they have not included in computing their MVP score. It is recommended that they add to the comparison method that consider the gene entity such as MSC and GAVIN. Moreover, The authors seem to include Eigen and PP2 though in computing MVP and then compare MVP against them which is obviously an unfair comparison. The authors experiment with removing groups of features from their predictor and, as expected, the deterioration in performance is worst when excluding the published predictors showing that MVP is harnessing the power of multiple predictors built on heterogeneous assumptions. To verify this it would be informative to show ROC curves when training exclusively on the published predictors without the additional features and excluding all predictors used to compare against from the training features (e.g., PP2 and Eigen).

4. Regarding evaluations, supplementary figure S3 showing ROC curves on the training dataset does seem to overfit the training set (AUC=0.99) and this could be due to improper tuning of the capacity (hyperparameters) of the neural network or the small size of the training set (and alternatively, an overlap between the training and validation sets). Please show how the results change when the training set size is increased by using both constrained and unconstrained gene sets together, and verify non-overlap between training and validation sets. As mentioned above, the negative training set could be "too easy", please include random variation from gnomAD exomes in the negative training set. It is also important to specify how the authors decided on the parameters for the NN and whether different values for them were tested.

5. I am not convinced what is the conclusion of the proposition to distinguish between constrained and unconstrained genes in training . An alternative to this is to simply have the criterion used to distinguish the two categories of genes (pLI) as a training feature which the authors already do anyway. It does seem that variation in constrained genes is easier to predict (expected, as variations in these genes are usually more damaging) but this could also be due to the fact that

the authors have selected rarer ($MAF=10^{-6}$) variants for the constrained genes compared to the unconstrained ones ($MAF=10^{-4}$). The authors also use more features (38 versus 21) when predicting constrained genes versus unconstrained ones introducing artificial differences between the two which might explain the better ROC curves for the former.

6. My biggest concern with this new method is the neural network architecture employed and why this is a CNN rather than a regular feed forward network given the small number of heterogenous features that are employed. The reason CNNs replaced feed forward networks on image recognition was mainly because of the unmanageable number of trainable parameters when every pixel in the image becomes a feature. The convolutional filters used by CNNs drastically reduce the number of parameters by allowing parameter sharing. Since this method employs a very small number of features and training examples by neural network standards, the authors should show how a standard feed forward neural network appropriately trained (number of layers, number of neurons) or a machine learning classifier such as Random Forest or SVM perform.

> Reviewer #1 (Remarks to the Author):

>

> In this manuscript, the authors describe a new deep learning method called MVP for predicting pathogenicity of missense variants. Based on several different data sets, the proposed method was shown to perform clearly better than the other methods considered in the comparisons. Whether the better performance can be attributed to the new method requires additional supporting data, as detailed below.

>

> Major comments:

> 1. The authors have focused on missense variants only. For many complex diseases, the statistically associated coding variants have been well-cataloged and/or they can only explain a certain (sometimes fairly small) fraction of disease susceptibility/heritability. The authors have chosen not to focus on the much larger set of non-coding variants and the more difficult problem of predicting effects of non-coding variants. Even for genic variants, the authors have chosen not to consider nonsense, frameshift and splicing variants. Considering that in disease genomic studies the significance of a coding variant is indicated by its relative frequencies in cases and controls, the primary use of the proposed method is in the study of rare variants. Given all these restrictions, the authors need to explain more clearly the value of their method in terms of the magnitude of increased accuracy as compared to other previous methods.

>

We agree that other types of variants also make significant contribution to disease genetics. We indeed have an ongoing project to predict the pathogenicity of noncoding variants, but the underlying biological model is entirely different and therefore it is out of scope of this manuscript. We argue that the problem of predicting pathogenicity of coding variants is still a major pressing issue in genetic studies and clinical genetic diagnosis, due to the availability of very large-scale exome sequencing data. Among coding variants, missense variants are uniquely difficult to analyze. In Supplementary Table 1, we show that only about 15% of de novo missense variants observed in these early onset diseases are truly pathogenic, and that published methods have limited performance. MVP can achieve better precision with higher recall.

We also agree that nonsense and other protein-truncating variants (PTVs) are very important, but the mechanism of pathogenicity of PTVs is very different to missense variants: PTVs usually lead to non-sense mediated decay that cause substantial reduction of gene expression level, whereas missense variants mostly change the function of the protein without direct affecting expression level. Therefore, whether a PTV is likely pathogenic largely hinges on whether the gene is dosage sensitive (i.e. haploinsufficient). In other words, it's a separate computational problem and beyond the scope of this study.

> 2. The manuscript does not include a clear definition of pathogenicity, but implicitly implies its meaning based on the data sets used (by adopting the definitions used by these data sets) or recurrence in cases and controls. Since pathogenic phenotype could be defined in various ways covering a wide spectrum of severity, it is important to explain which definition(s) is(are) used in this study and whether the data sets considered are consistent with this(these) definition(s).

>

We added the definition of pathogenicity used in this study in line 35.

Pathogenicity is defined at phenotypic level: that a variant cause disease. This is in contrast to deleteriousness at molecular level (affect protein function). But the latter is the foundation of the former. Consistent with the definition, we used “pathogenic” variants curated by clinical databases as positives for model training. Such pathogenicity is indeed correlated with recurrence in cases. We used cancer somatic mutation hotspots as one of the data sets for performance evaluation. Mutation hotspots were identified based on recurrence in cancer genomes.

> 3. The pathogenicity of a variant should depend highly on the particular disease of interest. For instance, if a variant disrupts an important function of a protein, but the protein is not expressed in the relevant cell or tissues types of a disease, it may not be pathogenic in the context of this disease. In the whole manuscript, pathogenicity is discussed in a way that does not take into account the disease context. This is exemplified by the training of a single model that was applied to several different data sets for very different diseases. Is it sufficient to construct a single general model for all diseases?

>

This is a great question that touches a fundamental issue in this field.

Mechanistically, a missense variant changes the function of a protein; if the protein is critical to a biological process, such functional changes may lead to disease(s); and if the disease(s) reduce reproductive fitness of the patient, the variant will be less likely to be transmitted to the next generation and therefore under negative selection in human population and “conserved” across species. It would be conceptually elegant to have separate predictions for the impact on protein function and impact on phenotypes. However, we have very incomplete knowledge about how protein function is changed by amino acid changes, and we have even less knowledge about the full list of genes that are involved in specific diseases. As a result, the most effective methods for predicting pathogenicity are all based on integration of information from conservation and protein function. Since conservation is essentially an integral over all possible phenotypic consequence, it is nearly impossible to tease out disease context if we use conservation information in prediction. In this context, just like previous methods, using a single general model for all diseases is a pragmatic solution in order to take advantage of conservation information.

There are ways to optimize prediction for specific genes or diseases. But to do that we need different models and methods that are beyond the scope of this manuscript.

> 4. The authors suggest that MVP performs better than the previous methods because of the methodology, including the use of deep residual network and the choice of input features. This could indeed be the case, but it is also possible that the method performed better due to the training data. There are two related issues here:

> A. Is it possible to train some of the previous methods using exactly the same training set? In particular, if the CNN method DANN is trained using this set of data, would it also perform as well as MVP? Without this controlled comparison, it is hard to evaluate the significance of the proposed deep residual network architecture.

> B. The authors have carefully removed variants in the testing sets that are also present in the training set. However, it is not clear whether some features directly contained information of the testing variants. In particular, since the predictions of many methods were collected from dbNSFP and they were used as input features to MVP, it is important to ensure that the testing data were not involved in training any of these methods.

>

For A: It is very hard to replicate all the published methods since many of them do not release the source code. In our study, the training data has been used in other methods: M-CAP and REVEL used HGMD variants as positives, MetaSVM used Uniprot database; there is overlap between HGMD and Uniprot positive variants. DANN used Fully-connected Neural Network as their architecture, while MVP is based on convolutional neural networks. To address the question how deep neural networks contribute to MVP's performance, we trained a Random Forest model (RF) and Fully-connected Neural Network (FCNN) using the same training datasets and the same features with MVP. MVP has better performance than RF and FCNN in prioritization of *de novo* variants. (Supplementary Fig S9, S10, S11).

For B: We have several testing data sets. The cancer hotspot data were identified from statistical evidence by a recent publication (Chang *et al* 2016); the *de novo* mutation data were from recent large-scale sequencing studies of autism and congenital heart diseases. Part of the cancer hotspot data that were identified in earlier publications (e.g. before 2014) could be used in training of the methods collected by dbNSFP. Nearly all of the *de novo* mutation data were published after these methods.

>

> 5. Related to #4a above, the authors have only cited a reference and given an abstract explanation (that deep residual network can better use big training data) for their choice of deep residual network over other machine learning methods. A more thorough discussion would be necessary to help explain the advantages of this

method and guide future research on similar topics.

>

We now expanded the explanation in the discussion part (line 195-221). Basically, a deep neural network has the capacity to model high-dimensional data with correlated features, and ResNet is efficient in training. We also compared our model architecture with other neural network architecture (Fully-connected Neural Network) and our methods shows better performance.(Supplementary Fig. S9-S11).

> 6. By predicting the pathogenicity of each variant independently, the authors are making an assumption that the disease phenotype does not depend on the interaction of multiple variants.

>

We acknowledge that interactions do exist among multiple variants. There are published methods that exploit co-evolution of pairs of residues to infer functional importance. A few recent papers describe methods to infer mutation hotspots by spatial proximity in 3D in proteins with known structure. It is possible to engineer predictive features based on the information from co-evolution or 3D spatial proximity, but the model will be substantially different to what we described in this manuscript. We do note that it is unlikely to observe two or more rare variants in the same gene in one human subject, and in human diseases, there are very few known genetic interactions that involve multiple rare variants.

> 7. While the authors have considered many types of features, it seems that they have not (at least directly) use features related to protein domains, functional annotations of genes and context-specific activities such as gene expression.

>

We agree that protein domain is informative, but we did not include it explicitly in the model for a few reasons: (1) Many known pathogenic variants are not located in known domains; (2) the functional importance of domains is partly captured by conservation scores which are already included in the model as features; (3) different domains have very different functions. We believe ultimately it is more helpful to use domain information in downstream genetic analysis where researchers have specific expertise in certain proteins and diseases. Functional annotations and gene expression are valuable data, but also it's better to use these data in downstream analysis in specific diseases.

> 8. In Supplementary Table 1, how are the score thresholds of the different methods determined? If the objective is to maximize the precision alone, usually it can be achieved by setting a higher score threshold to predict a smaller number of risk variants.

>

Supplementary Table 1 used the recommended threshold by the publications. The objective is usually not precision alone. In genetic studies aiming for finding new risk genes, the optimal method is the one with optimal area under the precision-recall curve (Zuk *et al* PNAS 2014). That is the reason we use inferred precision/recall for performance assessment by comparing *de novo* mutations between cases (autism or congenital heart disease) and controls. In typical clinical genetic testing, the objective is to maximize recall with high but approximately fixed precision (such as 0.95). The precision-recall curve (such as Figure 4) would be informative for this purpose as well.

> 9. To what extent is it confident that the random rare missense variants from population data can be considered negative training data?

>

This is a good question. Our answer is it depends on the purpose. The population data are from adult cohorts that mostly do not have developmental disorders or other severely early onset diseases. The vast majority of rare missense variants are inherited (for each person, there are about 0.5 *de novo* missense and 100 rare inherited missense variants). About 15-20% of rare *de novo* missense variants are pathogenic based on case-control comparison. The fraction of rare inherited missenses that are pathogenic is unknown. Previous studies of autism (Krumm, <https://www.nature.com/articles/ng.3303>) show that the cases and controls do not show significant difference in burden of inherited rare missense variants. This indicates that the fraction of pathogenic inherited missense is very low. Therefore, the vast majority of random missense variants in these subjects would not have large genetic effect that would be considered pathogenic. Because of this, it is reasonable to consider them as negatives in training.

To further reduce pathogenic variants in negative training, we previously had removed the small number of variants that are also present in the positives.

> 10. The negative data used in defining the testing sets need to be explained in more detail. This is important for evaluating whether there are some obvious differences between the positives and negatives.

>

In *de novo* datasets, the negatives variants are from 1911 unaffected siblings of Simons Simplex Collection, the data processing pipeline are similar between the positives in other WES studies and show no obvious batch effects and no enrichment in synonymous variants (Supplementary Table S10). This control dataset has been used in comparison in multiple WES studies (Iossifov *et al*, Nature 2014; Homsy *et al*, Science 2015; Qi *et al*, Human Mutation 2016)

In the cancer hotspot dataset, the negatives are randomly selected variants in the same gene set from population in DiscovEHR database. These negatives do not overlap with the negatives used in training.

> 11. In Figure S6a, it is seen that for constrained genes, removing the published predictors led to a fairly big decrease of the AUC value. What would be the AUC value with only these predictors, integrated by the proposed deep residual network or a simpler meta-learning method?

>

We observed worse performance when training the model with only published predictors. In cancer hotspot testing dataset, there is an AUC drop of 0.04 in constrained genes, a drop of 0.06 in non-constrained genes, indicating the important information from contribution of other features. See Fig.2, which is an updated version of previous Supplementary Fig. S6, with the performance of model with only published predictors added.

> 12. Most of the performance evaluations in this study are based on ranking of the predicted pathogenicity. This decision of the authors is reasonable, since the issue of finding a suitable score threshold for every method can be avoided. On the other hand, the absolute score could also carry important information. For instance, practically users of these scores may want to focus on the variants considered "severe" or "damaging" by the prediction method, rather than focusing on a fixed number of top variants. Did the authors find a particular score threshold/range of their method that seems suitable for the different data sets considered, or do they have a recommended way for tuning the threshold when analyzing an unseen data set?

>

In the manuscript, we recommend using a rank score of 0.7 (corresponding to raw value 0.15) in constrained genes and rank score 0.75 in non-constrained genes to define pathogenic, as such values achieved the best balance between sensitivity and specificity. See Supplementary Fig. S13 for the one to one transformation between rank score and raw score. We have clarified this point in the manuscript in line 147 and Supplementary Fig. S7.

> Minor comment:

>

> 1. In Figure S3, how can the mean of six 0.99 to be 0.98?

>

Fixed. See Figure S4.

>

> Reviewer #2 (Remarks to the Author):

>

> Qi et al. describes a new method for predicting pathogenic missense mutations

called MVP. Compared to previous methods, MVP uses deep residual neural network to leverage large training data sets and to deal with correlated co-variates. Using several large sets of annotated pathogenic missense mutations, the authors demonstrate MVP outperforms several state-of-the art methods. The use of deep neural network is innovative although all statistical features used in MVP have been tested before. Given the black-box nature of deep neural network, it is hard to figure out the underlying mechanisms that lead to the performance gain. Additionally, the following major concerns need to be addressed before the manuscript becomes suitable for publication.

>

> Major comments:

>

> 1. “We reason that the variants in constrained genes (ExAC pLI \geq 0.5) and non-constrained gene (ExAC pLI $<$ 0.5) may have different modes of action of pathogenicity”?

> 2. Can the authors provide more rationale and supporting evidence for this hypothesis? How would the performance results change if a difference pLI threshold is chosen?

>

This is an important point. By mode of action, there are three major types for missense variants: hypomorphic (partial loss of function), gain of function, or dominant negative (a heterozygous genotype will have similar loss of function consequence as a homozygous genotype). If a gene is non-constrained, i.e. it is not depleted of loss of function variants in the population, then *heterozygous* loss of function variants (such as stopgain) in this gene are unlikely to cause severe conditions, therefore, it's unlikely that a hypomorphic missense variant with *heterozygous* genotype in the gene can be pathogenic. On the other hand, gain of function or dominant negative missense variants with heterozygous genotypes can have an impact similar to homozygous variants, therefore, such variants in non-constrained genes can still be pathogenic. If a gene is constrained, i.e., depleted of loss of function variants in the population, then all three groups of variants with heterozygous genotype in the gene can be pathogenic. The vast majority of rare variants have heterozygous genotypes in patients. Therefore, rare pathogenic variants would have different modes of action in genes that are depleted with loss of function variants in general population (“constrained”) compared to the genes that not (“non-constrained”).

We use ExAC pLI (probability that a gene is intolerant to a Loss of Function) metric to define constrained genes (pLI \geq 0.5). In Lek et al 2015 and our own previous study (Han *et al* 2018), high pLI is well correlated with disease causing genes with a dominant model, i.e. through heterozygous pathogenic variants. This idea has been explored in a recent study on developmental disorders (Deciphering Developmental Disorders Study 2017, PMID: 28135719): researchers used a similar assumption as

ours to estimate the proportion of pathogenic missense mutations being loss of function (hypomorphic).

We added a brief description on such assumption in line 57 and line 72.

In terms of pLI threshold, we show the distribution of pLI scores of all genes in Supplementary Fig. S14. The distribution is nearly bimodal. 80% positive training variants are with $pLI > 0.9$ in constrained genes and 85% positive training variants are with $pLI < 0.1$ in non-constrained genes.

	Number of genes	Number of variants in positive training
$0.9 < pLI \leq 1.0$	3225	11190
$0.5 < pLI \leq 0.9$	2217	2459
$0.1 < pLI \leq 0.5$	2398	2482
$0 < pLI \leq 0.1$	10368	15310

We trained our model using use two different pLI cutoffs: on variants with $pLI > 0.9$ for constrained genes and $pLI < 0.1$ for non-constrained genes. We showed that among cancer hotspot dataset, the performance is very similar, indicated by AUC = 0.912 trained with $pLI > 0.5$, AUC = 0.911 trained with $pLI > 0.9$, AUC = 0.849 trained with $pLI < 0.5$, AUC = 0.846 trained with $pLI < 0.1$. (Supplementary Fig. S15) We reason that the majority of information comes from genes highly haploinsufficient or non-haploinsufficient, and we will need more variants between pLI 0.1 to 0.9 to increase the power.

> 3. A sensitivity analysis of the deep neural network model needs to be performed, i.e. how does the various parameter setting of the neural network affect its performance?

>

We added information on hyperparameters (line 348-356) in Methods. We tested models of different number of residual blocks and number of neurons in fully-connected layer. The model is sensitive to parameters in early layers. Deeper models lead to fast overfitting and unstable performance in testing datasets.

> 4. AUC curves are used for performance comparison. However, given the small difference in the AUC scores, are they statistically significant? For instance, in Figure 1, is the difference between MVP and REVEL statistically significant? And the difference between MVP and M-CAP and REVEL significant? Same question for the supplemental figures.

>

Yes. For example, when using methods (DeLong, et. al., 1988) to compare paired ROC curves, the difference of AUC for cancer hotspots data in MVP and REVEL (which is the second best) is $3.17e-8$ and $6.6e-4$, respectively in constrained and non-constrained genes. We added the maximum p value in line 115.

> 5. There are a few issues with Figure 3. Figure 3D shows MPC outperforms MVP in ASD constrained genes. Any explanation? In Figure 3E, why there is no curve for MPC? In Figure 3C for all genes, doesn't MPC achieves better sensitivity at the published recommended threshold (indicated by a star) compared to MVP (threshold = 0.7)?

>

We acknowledge that MPC has better precision with constrained genes with autism data. We do note MVP has much better performance in constrained genes with CHD data, and much better performance in non-constrained genes with data from both conditions. Based on known risk genes (SFARI gene for autism, Jin et al 2017 curated for CHD), autism risk genes are under stronger negative selection of loss of function variants (Cassa et al 2015) than CHD risk genes. High score of sub-genic regional constraint, a key metric used in MPC, is predominantly located in genes under very strong negative selection. This could explain the difference of autism vs CHD mutations among constrained genes. Supplementary Fig. S18 shows that variants in constrained genes significantly have higher MPC scores than those in non-constrained genes, either for pathogenic variants or benign variants in the training set.

MPC has limited performance among non-constrained genes, the results of all threshold in MPC in non-constrained genes is not statistically significant in ASD cases, so we did show MPC in the panel on non-constrained genes.

There is a trade-off between sensitivity and specificity. The recommended threshold of MPC achieves slightly better sensitivity compared to recommended threshold of MVP, however, the specificity of MVP is almost doubling MPC. With the same sensitivity, MVP generally outperformed other methods.

> 6. The manuscript is written without enough details. For instance, how are previous methods run in the benchmarking analysis? Using default parameters? The Discussion section is not really a discussion. It should be expanded to discuss issues such as 1) what is the reason for the performance gain by the deep neural network approach; 2) Additional candidate pathogenic mutations are predicted for CHD. But what are they, any new biological insights?

>

We used standard annotation tool (dbNSFP) to obtain predicted scores for all published methods. We did not re-train previously published methods. Some of the previously published methods may have used part of the Varibench or cancer mutation hotspots data in training. That issue, if exist, would favor the performance of our method.

We acknowledge the discussion section lacks details. We now have expanded it to include discussions about the performance gain and the biological insights from the analysis of CHD data (line 195-221 and 223-239). The performance gain by the deep neural network can be attributed to two aspects. First, we included correlated features at several different levels e.g. conservation, mutation intolerance and published methods. We have showed that only use a subset of those features cannot achieve the optimal performance as using all the features (Fig. 2). Second, it is well known that deep neural network has a larger model capacity to leverage large training data sets for improving prediction. In this revision, we show that using the same features and training dataset, convolutional neural networks can achieve better performance than Random Forest and Fully-connected Neural Network (Supplementary Fig. S9-S11).

In the analysis of CHD data, we observed that de novo mutations contribute much more to isolated CHD cases than previous estimates. This finding is related to better understanding of the genetic architecture. To gain more biological insights, we performed pathway enrichment analysis of predicted pathogenic missense variants using Enrichr (Kuleshov et al 2016). We show that genes with predicted pathogenic variants in isolated CHD cases are significantly enriched in muscle contraction and cardiac conduction, two pathways that do not show marginal significance in syndromic cases which have extracardiac disorders (Supplementary Figure S12). In contrast, the genes with predicted pathogenic variants in syndromic CHD cases are enriched in Notch, Robo, or MAPK signaling pathways, all of which have been implicated in other birth defects and developmental disorders. This finding is consistent with the phenotypes of these CHD patients.

>

>

> Reviewer #3 (Remarks to the Author):

>

> The manuscript “MVP: predicting pathogenicity of missense variants by deep learning” describes a new method to predict pathogenicity of missense variants using a deep learning approach, with convolutional neural network (CNN) called a residual network (ResNet). The study is timely and the effort of applying cutting-edge machine learning approaches in pathogenicity prediction are important.

However, I have several major concerns regarding the study design and the results.

>

> 1. As a neutral/benign dataset, the authors could use a set of random variants

from the human population (such as from gnomAD), since such set will contain by definition a vast proportion of variants that are not pathogenic and represent the average human exome.

>

This is indeed similar to what we did: we used random rare variants from DiscovEHR (Dewey et al 2016) and two other data sets (UniProt negatives and Human-derived changes from the CADD database) as negatives. The DiscovEHR data is from a large population similar to gnomAD. The only reason we did not use random variants from gnomAD as negatives is that we used gnomAD allele frequency to define rare variants. In order to avoid double dipping of the data, we chose an independent data set, DiscovEHR, for selecting random variants observed in the population.

> 2. The negative training set consists of rare variants from DiscovEHR, Uniprot negatives and Human-derived changes from the CADD database. It is important that the authors specify what is the MAF range and CADD scores of mutations employed from each of these three datasets. The concern is that the authors are comparing highly pathogenic mutations against trivially benign variants which are excluded as a first step of data filtering in any WES study. The other concern is training on CADD database mutations when CADD itself is one of the training features.

>

We completely agree that trivially benign variants should not be considered here. In the training and testing, we only include rare variants (allele frequency < 1% gnomAD dataset). This approach has been shown to be effective by previous methods (REVEL and M-CAP). We showed the distribution of CADD scores in the training data set in Supplementary Fig. S17, but we did not use CADD scores as a filter to set training data. Besides, CADD itself is not a training feature in our model. (Supplementary Table S2)

> 3. The authors perform thorough evaluations and show very good performance of their method when comparing it with 13 other pathogenicity prediction tools that they have not included in computing their MVP score. It is recommended that they add to the comparison method that consider the gene entity such as MSC and GAVIN. Moreover, the authors seem to include Eigen and PP2 though in computing MVP and then compare MVP against them which is obviously an unfair comparison. The authors experiment with removing groups of features from their predictor and, as expected, the deterioration in performance is worst when excluding the published predictors showing that MVP is harnessing the power of multiple predictors built on heterogeneous assumptions. To verify this it would be informative to show ROC curves when training exclusively on the published predictors without the additional features and excluding all predictors used to compare against from the training

features (e.g., PP2 and Eigen).

>

We used independent dataset from cancer hotspot and *de novo* dataset for testing, using such dataset are fair to assess the performance of different methods since none of them used the testing dataset to tune the model. Including scores such as Eigen and PP2 in the comparison provides a baseline performance and shows that using additional features and deep learning methods can improve the performance. The superior performance of MVP over Eigen and PP2 is expected.

We showed worse performance when training the model with only published predictors compared to the full model. In cancer hotspot testing dataset, there is an AUC drop of 0.04 in constrained genes, a drop of 0.06 in non-constrained genes, indicating the important information from contribution of other features. See Fig. 2.

As for the comparison with GAVIN(<https://molgenis20.gcc.rug.nl/>), it gives binary outcome prediction with only 3498 genes, thus preventing a genome-wide comparison. MSC(<http://pec630.rockefeller.edu:8080/MS/>) also gives binary cutoff for prediction which makes it hard to vary the threshold and compared with other methods by ROC.

> 4. Regarding evaluations, supplementary figure S3 showing ROC curves on the training dataset does seem to overfit the training set (AUC=0.99) and this could be due to improper tuning of the capacity (hyperparameters) of the neural network or the small size of the training set (and alternatively, an overlap between the training and validation sets). Please show how the results change when the training set size is increased by using both constrained and unconstrained gene sets together, and verify non-overlap between training and validation sets. As mentioned above, the negative training set could be “too easy”, please include random variation from gnomAD exomes in the negative training set. It is also important to specify how the authors decided on the parameters for the NN and whether different values for them were tested.

>

We verified that there is no overlap at all between training and validation sets. But we agree that the extremely high AUC value shown in S3 demands caution. In deed we discussed the potential problem in the supplementary notes “performance inflation in different datasets”. While overfitting is always a concern with machine learning methods with many parameters, we believe the biggest factor here is how the training data and testing data sets were curated: both HGMD (used in training) and Varibench (used in testing) databases of pathogenic variants were curated from the literature. There are likely similar factors causing false positives in such process across different databases. We alluded to this issue in line 108 and Supplementary notes. For that reason, we used the cancer hotspot datasets and *de novo* mutation dataset as the main approach for evaluating and comparing performance. We also

use cancer hotspot data to explicitly estimate the inflation of performance assessed by Varibench data. (Supplementary Fig S6)

We have three sources of negative training, DiscovEHR, Uniprot negatives and Human-derived changes from the CADD database. We did not include random variants from gnomAD database in training or testing for two reasons. We used gnomAD allele frequency as a feature. The M-CAP model used Exome Aggregation Consortium data set version 0.3 (part of gnomAD database) and did not reported the variants used in training which makes it hard to exclude those variants in testing. For these two reasons, we need an independent dataset such as DiscovEHR dataset as well as *de novo* dataset from SSC for testing.

We now expanded the Methods to describe the hyperparameters (line 348-356). In the MVP neural network, we tested different number of residual blocks. With all other parameters fixed, the model with two residual blocks contain 12,544 parameters before fully connected layers, and it saturates at around 20 iterations. Adding another residual block will increase the parameter number to 18,752 and the model saturates at around 8 iterations, which indicates quick overfitting. Larger and cleaner datasets are needed for training a deeper network to fully utilize its power. We also tried up to 1024 neurons in the first fully connected layer, which doubles the parameter number of 512 neurons and makes the model overfit soon. Other hyperparameters we chose are commonly used in deep learning models, including kernel size of 3, pooling size of 2, depths of 32 and ReLU as activation functions.

> 5. I am not convinced what is the conclusion of the proposition to distinguish between constrained and unconstrained genes in training. An alternative to this is to simply have the criterion used to distinguish the two categories of genes (pLI) as a training feature which the authors already do anyway. It does seem that variation in constrained genes is easier to predict (expected, as variations in these genes are usually more damaging) but this could also be due to the fact that the authors have selected rarer (MAF=10⁻⁶) variants for the constrained genes compared to the unconstrained ones (MAF=10⁻⁴). The authors also use more features (38 versus 21) when predicting constrained genes versus unconstrained ones introducing artificial differences between the two which might explain the better ROC curves for the former.

>

There are two reasons we trained the model for constrained genes and non-constrained genes separately.

(1) The mode of action in constrained genes is different in constrained genes and non-constrained genes as described in response to Reviewer #2. We used pLI (probability that a gene is intolerant to a Loss of Function) metrics to separate constrained genes and non-constrained genes. pLI has a good correlation with haploinsufficiency (Lek et al, Nature 2016), therefore hypomorphic, gain of function,

and dominant negative missense can all be pathogenic in these genes. In genes that are not haploinsufficient, hypomorphic cannot be pathogenic and genes with pLI < 0.5 are mostly not haploinsufficient. In a DDD study (<https://www.nature.com/articles/nature21062>), researchers found significant enrichment of missense variants in both dominant haploinsufficient genes and non-haploinsufficient genes with known altered-function and loss-of-function DD-associated genes.

(2) Empirically, previously published methods have poor performance in detecting pathogenic variants among non-constrained genes; we therefore removed most published methods in training for non-constrained gene to increase the weight on protein structure features and to reduce number of parameters in the model. Models with fewer parameters are more robust to overfitting.

We showed the results of train the model combined both constrained genes and non-constrained genes, the testing performance is similar among constrained genes but worse in non-constrained genes (Supplementary Figure S16).

The performance for constrained genes with (MAF=10⁻⁴) is similar with MAF 10⁻⁶ (data not shown), and have larger enrichment and more significant than non-constrained genes. We used the threshold of 10⁻⁶ since pathogenic variants in constrained genes are usually under stronger selection and consequently rarer in population, as shown in a recent publication (Kosmicki et al, Nature Genetics 2017).

> 6. My biggest concern with this new method is the neural network architecture employed and why this is a CNN rather than a regular feed forward network given the small number of heterogenous features that are employed. The reason CNNs replaced feed forward networks on image recognition was mainly because of the unmanageable number of trainable parameters when every pixel in the image becomes a feature. The convolutional filters used by CNNs drastically reduce the number of parameters by allowing parameter sharing. Since this method employs a very small number of features and training examples by neural network standards, the authors should show how a standard feed forward neural network appropriately trained (number of layers, number of neurons) or a machine learning classifier such as Random Forest or SVM perform.

This is a good question. One of the main reasons we choose CNN over regular feed forward network is to reduce the number of parameters in the model. Even in a simple 3-layer network, a fully connected layer with 256*256*256 neurons will have 637,534,208 parameters. Given the limited number of training dataset, it can quickly go to overfitting and result in large fluctuation in performance. In the CNN framework, there are 12,416 parameters in the residual layers and total 636,161 parameters in constrained model and 357,633 parameters in non-constrained model. In our CNN model setting, we put highly correlated features closely so that first residual layers can capture local context interaction within groups while high order

residual layers can capture non-linear interaction between groups. To further answer this question, we trained a fully-connected Neural Network (FCNN) model and Random Forest (RF) model with same features and training datasets. The CNN-based method outperforms FCNN and RF on de novo variants (Fig. S9-S11). We acknowledge that larger and cleaner data set is needed to train a deeper network to fully utilize the power of deep learning model.

Reviewer #1 (Remarks to the Author):

The authors have provided detailed point-by-point responses to my comments in the first round of review. Based on their responses, here I provide follow-up comments that the authors need to address.

Original major comment 1: The authors emphasized on the importance of missense mutations and the difficulties in predicting their pathogenicity. While I agree with them on these points, they need to show that models specifically trained for missense mutations are better than the models for pathogenic genetic variants in general. For instance, since the first submission of this manuscript two years ago, quite a number of papers about predicting the functional effects or pathogenicity of genetic variants using various artificial neural network-based methods have been published or deposited into preprint servers such as bioRxiv. Although the authors have already compared their proposed method with many other predictors, updating their comparisons with the latest methods included is necessary given the long duration since the submission of the first version of the manuscript.

Original major comment 2: The authors have now clearly defined that a pathogenic genetic variant is one that causes disease. A practical difficulty in using this definition is that for some, perhaps most, data sets, there are only association signals between the genetic variants but not direct proofs of their causal effects on diseases. In addition, the authors' definition also does not explain whether the genetic variant by itself should be either sufficient or necessary for the disease. These comments are not merely for a conceptual discussion, but are also important for understanding what the models are designed for. For instance, identifying variants with large effect sizes could require very different features and models from those for identifying variants with moderate effect sizes.

Original major comment 3: Based on the authors' explanations, I would classify their goal as predicting the potential for a genetic variant to be pathogenic. A genetic variant with a low pathogenic potential would not be pathogenic in any context, but even one with a high pathogenic potential, it would still be pathogenic only in particular contexts. Predicting pathogenic potential seems to me an older problem, while newer methods have started to incorporate contextual information. One typical approach is to first learn the potential or a general embedding of the input features, and then use it as the input of a second-stage context-specific model for each particular context such as a specific disease. This second-stage model may take additional context-specific features as input, or it may simply take the learned potential scores/embedding as the only input and find a way to use the information for the specific context. To make the current manuscript more inline with this new trend, the authors should at least use one data set to investigate how their pathogenic scores are used in different context-specific models.

Original major comment 4: Based on the new results provided by the authors (Figure S9 and the ASD results in Figure 11), it is actually unclear to what extent the superior performance of MVP was due to its residual network architecture as compared to the set of features included. In fact, both can be important contributions of this work, but if the feature set indeed plays a key role in the prediction performance regardless of the model, additional analyses are required to explain how this feature set differs from the ones used in previous studies and why it can perform better.

Original major comment 5: The authors have now provided some explanations for their use of the residual network architecture in the Discussion section (the line numbers given by the authors seem incorrect), but if the authors think this network architecture is fundamental to the whole work, they should justify it early in the manuscript instead. In particular, they need to explain 1) the characteristics of convolutional neural networks, 2) the advantages of the residual variation over standard CNNs, and 3) the differences between the residual network and other popular architectures.

Original major comment 6: I agree with the authors that if only rare variants are considered, it is not common to find multiple rare variants in the same gene. However, the authors have not actually restricted the scope of their study explicitly to only rare variants. They should clarify it at the beginning of the manuscript if it is crucial to the whole study.

Original major comment 7: I have commented on the context-specific features in the comment about two-stage modeling above. As for protein domain features, they can at least provide a strong prior to the predictions, and thus if they are really not useful or their information is already contained by other features, evidence should be provided. An additional related comment here is that the authors used various engineered features but not the raw DNA/protein sequences around the genetic variant, a decision that seems to be contradictory to the spirit of CNNs and may not have fully utilized the capabilities of CNNs.

Reviewer #2 (Remarks to the Author):

Thank you for your effort addressing my questions. The follow two issues remain to be addressed:

Comment #3. A sensitivity analysis of the deep neural network model needs to be performed, i.e. how does the various parameter setting of the neural network affect its performance?

Response: We added information on hyperparameters (line 348-356) in Methods. We tested models of different number of residual blocks and number of neurons in fully connected layer. The model is sensitive to parameters in early layers. Deeper models lead to fast overfitting and unstable performance in testing datasets.

Result of this analysis needs to be provided as supplemental figure.

Comment #5. There are a few issues with Figure 3. Figure 3D shows MPC outperforms MVP in ASD constrained genes. Any explanation? In Figure 3E, why there is no curve for MPC? In Figure 3C for all genes, doesn't MPC achieves better sensitivity at the published recommended threshold (indicated by a star) compared to MVP (threshold = 0.7)?

The following explanation needs to be incorporated into the manuscript.

"We acknowledge that MPC has better precision with constrained genes with autism data. We do note MVP has much better performance in constrained genes with CHD data, and much better performance in non-constrained genes with data from both conditions. Based on known risk genes (SFARI gene for autism, Jin et al 2017 curated for CHD), autism risk genes are under stronger negative selection of loss of function variants (Cassa et al 2015) than CHD risk genes. High score of sub-genic regional constraint, a key metric used in MPC, is predominantly located in genes under very strong negative selection. This could explain the difference of autism vs CHD mutations among constrained genes. Supplementary Fig. S18 shows that variants in constrained genes significantly have higher MPC scores than those in non-constrained genes, either for pathogenic variants or benign variants in the training set.

MPC has limited performance among non-constrained genes, the results of all threshold in MPC in non-constrained genes is not statistically significant in ASD cases, so we did show MPC in the panel on non-constrained genes.

There is a trade-off between sensitivity and specificity. The recommended threshold of MPC achieves slightly better sensitivity compared to recommended threshold of MVP, however, the specificity of MVP is almost doubling MPC. With the same sensitivity, MVP generally outperformed other methods."

Reviewer #3 (Remarks to the Author):

The authors have addressed all my concerns I have no further comments.

Reviewer #1 (Remarks to the Author):

The authors have provided detailed point-by-point responses to my comments in the first round of review. Based on their responses, here I provide follow-up comments that the authors need to address.

Original major comment 1: The authors emphasized on the importance of missense mutations and the difficulties in predicting their pathogenicity. While I agree with them on these points, they need to show that models specifically trained for missense mutations are better than the models for pathogenic genic variants in general. For instance, since the first submission of this manuscript two years ago, quite a number of papers about predicting the functional effects or pathogenicity of genetic variants using various artificial neural network-based methods have been published or deposited into preprint servers such as bioRxiv. Although the authors have already compared their proposed method with many other predictors, updating their comparisons with the latest methods included is necessary given the long duration since the submission of the first version of the manuscript.

In principle, coding and noncoding variants have different molecular mechanisms of pathogenicity. Coding variants, especially missense variants, may disrupt or change the function of a protein, whereas noncoding variants primarily disrupt the regulation of expression at mRNA or protein level. In this work, we are primarily interested in predicting the pathogenicity of missenses variants. We optimized the method for missense variants. To train both types of variants in one method is possible, but it is really beyond the scope of this work. We do note CADD, which was included in our comparison, was trained for both coding and non-coding variants.

We do agree that a few interesting methods have been published since our initial submission. To update the comparison, we now included two neural network-based methods, PrimateAI (Sundaram et al 2018) and UNEECON (Huang et al 2020), in the revision. Our method achieved overall better performance than these two methods with cancer hotspot data (revised Figure 1, Figure S6, Figure S7, and Table S4), Varibench data (Figure S5, Figure S6, and Table S4), and *de novo* variants from genetic studies of CHD and ASD (revised Figure 3, Figure 4, Figure S8, Table S1, Table S6, Table S7, and Table S8). We note that PrimateAI has good performance with *de novo* variants among constrained genes ($pLI \geq 0.5$) in ASD data (Figure 4D), but performs poorly in non-constrained genes (Figure 4E).

Original major comment 2: The authors have now clearly defined that a pathogenic genetic variant is one that causes disease. A practical difficulty in using this definition is that for some, perhaps most, data sets, there are only association signals between the genetic variants but not direct proofs of their causal effects on diseases. In addition, the authors' definition also does not explain whether the genetic variant by itself should be either sufficient or necessary for the disease. These comments are not merely for a conceptual discussion, but are also important for understanding what the models are designed for. For instance, identifying variants with large effect sizes could require very different features and models from those for identifying variants with moderate effect sizes.

We agree that pathogenic variants may have a range of effect size and penetrance. In this work, we focused on rare variants with large effect size. Our model is consistent with a common definition of pathogenicity (MacArthur et al 2014), that is, pathogenic variants

mechanistically contribute to disease, but not necessarily fully penetrance. Specifically, we included prediction features based on protein structure and protein modification, which provide biophysical and biochemical basis for mechanistical contribution to the disease, and evolution conservation, which is usually a consequence of direct contribution to diseases (instead of indirect contribution through linkage disequilibrium). Additionally, the positives in model training were from HGMD and UniProt, expert-curated databases with large-number of likely pathogenic and rare variants reported in previous publications. We did not include disease-associated common variants from genome-wide association studies (GWAS) in training.

This definition does not imply that a pathogenic variant is either sufficient or necessary to cause a disease: if a variant does not have full penetrance, it is not sufficient to cause disease; if the disease can be caused by other variants or genes, then it is not necessary. In genetic studies, pathogenicity is usually inferred based on association when the associated variant is extremely rare and has large effect size (e.g. relative risk or odds ratio), and ultimately confirmed by animal models. The goal of our work is to facilitate genetic studies of rare variants.

We have added the points in the discussion of the revised manuscript (line 295-306).

Original major comment 3: Based on the authors' explanations, I would classify their goal as predicting the potential for a genetic variant to be pathogenic. A genetic variant with a low pathogenic potential would not be pathogenic in any context, but even one with a high pathogenic potential, it would still be pathogenic only in particular contexts. Predicting pathogenic potential seems to me an older problem, while newer methods have started to incorporate contextual information. One typical approach is to first learn the potential or a general embedding of the input features, and then use it as the input of a second-stage context-specific model for each particular context such as a specific disease. This second-stage model may take additional context-specific features as input, or it may simply take the learned potential scores/embedding as the only input and find a way to use the information for the specific context. To make the current manuscript more inline with this new trend, the authors should at least use one data set to investigate how their pathogenic scores are used in different context-specific models.

We agree that predicted pathogenic variants would not be pathogenic in all contexts. This is indeed implicitly assumed in how the prediction methods, including MVP, were trained and how they are used in practice: (a) the positives used in model training were curated from publications where the likely pathogenic variants were implicated with one or a few conditions. (b) In genetic analysis, pathogenicity prediction is usually used in a condition specific context: either to prioritize variants a risk gene known to be implicated in a certain condition or searching for new risk genes of a certain condition. Therefore, it is generally not a problem. However, context-specific prediction can help with genes that have different modes of action (mechanisms) in different conditions. One example is *CTNNB1* (beta catenin). It's an oncogene in cancer through gain-of-function mutations (mostly somatic), and a risk gene in structural birth defects and neurodevelopmental disorders through loss of function germline mutations. Most prediction tools, including MVP, do not distinguish pathogenic missense variants with gain of function from the ones with loss of function. Explicitly predicting gain or loss of function would improve the utility of prediction methods. We do think this is an important question, and a two-stage model can be a part of the solution. This is however beyond the scope of this work.

While predicting pathogenic potential is not a new problem, but it is far from being solved, as shown in our analysis of *de novo* mutations.

Original major comment 4: Based on the new results provided by the authors (Figure S9 and the ASD results in Figure 11), it is actually unclear to what extent the superior performance of MVP was due to its residual network architecture as compared to the set of features included. In fact, both can be important contributions of this work, but if the feature set indeed plays a key role in the prediction performance regardless of the model, additional analyses are required to explain how this feature set differs from the ones used in previous studies and why it can perform better.

MVP achieved overall better performance than dense neural network (DNN) and random forest (RF) (Figure S9, S10, and S11) trained on the same training samples using the same prediction feature set. On cancer hotspot data, MVP is better than RF and achieved similar performance with DNN (Figure S9). And on *de novo* variants from genetic studies on ASD and CHD, if measured using *p-value*, MVP achieved better performance than both DNN and RF (Figure S10). And if measured using precision-recall-like curve, MVP achieved better performance for most of the selected thresholds. Specifically, in CHD data, MVP is much better than other methods. And in ASD data, MVP is better than other methods in high precision range, but similar in low precision range (Figure S11).

We totally agreed that both model architecture and feature set are important for the prediction performance. First, to investigate the relative importance of features, we fixed the model architecture and training samples, and we trained different models using different feature sets (Figure 2). Second, to investigate the contribution of model architecture, we fixed the feature set and training samples, and we trained a DNN model and a RF model (Figure S9, S10, and S11). We also note that some of the features were not included in previous methods such as the gene mutation intolerance features *s_het* and domino score. The sub-genic coding constraint was included by MPC but not by other existing methods. We show that the gene mutation intolerance features contribute a 0.015 and 0.017 of AUC improvement for constrained genes and non-constrained genes, respectively, with cancer hotspot data (Figure 2).

Original major comment 5: The authors have now provided some explanations for their use of the residual network architecture in the Discussion section (the line numbers given by the authors seem incorrect), but if the authors think this network architecture is fundamental to the whole work, they should justify it early in the manuscript instead. In particular, they need to explain 1) the characteristics of convolutional neural networks, 2) the advantages of the residual variation over standard CNNs, and 3) the differences between the residual network and other popular architectures.

The line numbers changed after we expanded our manuscript. We apologize for the oversight. The explanations are on line 221-229. We now also justified the rationale of our model early in the manuscript (line 91-97).

ResNet has proven very successful in computer vision²², structural bioinformatics²³, and the modelling of DNA/protein sequence motifs²⁴. First, convolutional layers in ResNet are capable of extracting hierarchical features or nonlinear spatial patterns from images or sequence data. Adjacent pixels in images and adjacent nucleotides/amino acids in DNA/protein sequences are highly correlated and local patterns emerge which can be efficiently captured by convolutional layers. To take advantage of this characteristic, we ordered the predictors based on their correlation, as highly correlated predictors are clustered together (Supplementary Fig. S2). Second, the residual component in ResNet can efficiently mitigate the issue of vanishing/exploding gradients and makes the training very efficient²². Third, compared with dense neural network, another widely used network architecture, the blocks of convolutional layers in ResNet can significantly reduce the number of parameters and can mitigate the overfitting problem.

We now added a brief description of these ideas in the revised manuscript (line 91-97).

Original major comment 6: I agree with the authors that if only rare variants are considered, it is not common to find multiple rare variants in the same gene. However, the authors have not actually restricted the scope of their study explicitly to only rare variants. They should clarify it at the beginning of the manuscript if it is crucial to the whole study.

We have clarified this early of this manuscript (line 103).

Original major comment 7: (A) I have commented on the context-specific features in the comment about two-stage modeling above. As for protein domain features, they can at least provide a strong prior to the predictions, and thus if they are really not useful or their information is already contained by other features, evidence should be provided. (B) An additional related comment here is that the authors used various engineered features but not the raw DNA/protein sequences around the genetic variant, a decision that seems to be contradictory to the spirit of CNNs and may not have fully utilized the capabilities of CNNs.

For A, as we mentioned in the response to the comment 3, we agree that two-stage modelling is a good idea to learn a disease-specific model and we also agree that domain feature is informative for disease-specific or other context-specific analysis. But it's debatable how much improvement we can achieve by including domain information in the model. First, the domain definition is not accurate for all proteins. The domains can be defined based on sequence or 3D structure. We do not have precise 3D structure for most of human proteins. Second, domains are correlated with sequence conservation and secondary structure, both already included as predictors in our model. For these reasons, we don't include domain features explicitly in our model.

For B, CNN has been widely used to learn patterns from raw DNA/protein sequences. However, its utility goes beyond learning patterns from raw data. In fact, our application meets the spirit of CNN. First, convolutional layers are usually for positional data such images and sequences, and we have extended its utility to tabular data. Adjacent pixels for images and adjacent nucleotides/amino acids for DNA/protein sequences are highly correlated and local patterns emerges which can be captured by convolutional layers. For our purpose, we first ordered the tabular features by the correlation such that adjacent features are more correlated

where convolutional layers can be used (Figure S2). Second, compared with dense neural network (DNN), CNN can significantly reduce the number of model parameters and can mitigate the overfitting problem. DNN can be regarded as a special case of CNN where the kernel size equals the size of the whole inputs instead of a local part. We have shown that our model can achieved overall better performance than DNN trained using the same training samples and feature set (Figure S9, S10, and S11).

We now added a brief description of these ideas in the revised manuscript (line 91-97).

Reviewer #2 (Remarks to the Author):

Thank you for your effort addressing my questions. The follow two issues remain to be addressed:

Comment #3. A sensitivity analysis of the deep neural network model needs to be performed, i.e. how does the various parameter setting of the neural network affect its performance?

Response: We added information on hyperparameters (line 348-356) in Methods. We tested models of different number of residual blocks and number of neurons in fully connected layer. The model is sensitive to parameters in early layers. Deeper models lead to fast overfitting and unstable performance in testing datasets.

Result of this analysis needs to be provided as supplemental figure.

We now have inserted Figure S19 to summarize the analysis. We used AUROC on cancer hotspot data to investigate the effects of the number of residual blocks on the testing performance. When 8 and 16 residual blocks are used, the AUROC for constrained genes are 0.885 and 0.872, respectively, and the AUROC for non-constrained genes are 0.822 and 0.814, respectively. We can see that due to limited training samples, deeper models will suffer the overfitting problem.

We also added this point in the revised manuscript (line 428-433).

Comment #5. There are a few issues with Figure 3. Figure 3D shows MPC outperforms MVP in ASD constrained genes. Any explanation? In Figure 3E, why there is no curve for MPC? In Figure 3C for all genes, doesn't MPC achieves better sensitivity at the published recommended threshold (indicated by a star) compared to MVP (threshold = 0.7)? The following explanation needs to be incorporated into the manuscript.

"We acknowledge that MPC has better precision with constrained genes with autism data. We do note MVP has much better performance in constrained genes with CHD data, and much better performance in non-constrained genes with data from both conditions. Based on known risk genes (SFARI gene for autism, Jin et al 2017 curated for CHD), autism risk genes are under stronger negative selection of loss of function variants (Cassa et al 2017) than CHD risk genes. High score of sub-genic regional constraint, a key metric used in MPC, is predominantly located in genes under very strong negative selection. This could explain the difference of autism vs CHD

mutations among constrained genes. Supplementary Fig. S13 shows that variants in constrained genes significantly have higher MPC scores than those in non-constrained genes, either for pathogenic variants or benign variants in the training set.

MPC has limited performance among non-constrained genes, the results of all threshold in MPC in non-constrained genes is not statistically significant in ASD cases, so we did show MPC in the panel on non-constrained genes.

There is a trade-off between sensitivity and specificity. The recommended threshold of MPC achieves slightly better sensitivity compared to recommended threshold of MVP, however, the specificity of MVP is almost doubling MPC. With the same sensitivity, MVP generally outperformed other methods."

As suggested, we now have added this explanation in the revised manuscript (line 273-293).

Reviewer #3 (Remarks to the Author):

The authors have addressed all my concerns I have no further comments.

Reviewer #1 (Remarks to the Author):

The authors have satisfactorily addressed my comments. I would just like to ask the authors to add some discussions related to Comment 3 (pathogenic potential versus actual pathogenicity in a particular context), which I believe would be helpful for some readers.

Reviewer #1 (Remarks to the Author):

The authors have satisfactorily addressed my comments. I would just like to ask the authors to add some discussions related to Comment 3 (pathogenic potential versus actual pathogenicity in a particular context), which I believe would be helpful for some readers.

We have added this point to the final part of the Discussion section (line 322-331):

Finally, we note that a single pathogenic score cannot capture the complexity of the mechanisms of pathogenicity. Many genes may have different modes of action in different diseases. One example is *CTNFB1* (beta catenin). It is an oncogene in cancer through gain-of-function mutations, and a risk gene in structural birth defects and neurodevelopmental disorders through loss of function germline mutations. Most prediction tools, including MVP, do not distinguish pathogenic missense variants with gain of function from the ones with loss of function. Explicitly predicting gain or loss of function, as what a recent study focused on channels did, would improve the utility of prediction methods.